# Selenium Alleviates the Adverse Effect of Drought in Oilseed Crops Camelina (*Camelina sativa* L.) and Canola (*Brassica napus* L.)

**DOI:** 10.3390/molecules26061699

**Published:** 2021-03-18

**Authors:** Zahoor Ahmad, Shazia Anjum, Milan Skalicky, Ejaz Ahmad Waraich, Rana Muhammad Sabir Tariq, Muhammad Ashar Ayub, Akbar Hossain, Mohamed M. Hassan, Marian Brestic, Mohammad Sohidul Islam, Muhammad Habib-Ur-Rahman, Allah Wasaya, Muhammad Aamir Iqbal, Ayman EL Sabagh

**Affiliations:** 1Department of Botany, University of Central Punjab, Bahawalpur Campus, Bahawalpur 63100, Pakistan; zahoorahmadbwp@gmail.com; 2Cholistan Institute of Desert Studies, The Islamia University of Bahawalpur, Bahawalpur 63100, Pakistan; shahzia.anjum@iub.edu.pk; 3Department of Botany and Plant Physiology, Faculty of Agrobiology, Food and Natural Resources, Czech University of Life Sciences Prague, 165 00 Prague, Czech Republic; skalicky@af.czu.cz; 4Department of Agronomy, University of Agriculture Faisalabad, Punjab 38000, Pakistan; uaf_ewarraich@yahoo.com (E.A.W.); marian.brestic@uniag.sk (M.B.); 5Department of Agriculture and Agribusiness Management, University of Karachi, Karachi 75270, Pakistan; sabir_tariq@yahoo.com; 6Institute of Soil and Environmental Sciences, University of Agriculture, Faisalabad 78000, Pakistan; muhammadasharayub@gmail.com; 7Department of Agronomy, Bangladesh Wheat and Maize Research Institute, Dinajpur 5200, Bangladesh; akbarhossainwrc@gmail.com; 8Department of Biology, College of Science, Taif University, P.O. Box 11099, Taif 21944, Saudi Arabia; m.khyate@tu.edu.sa; 9Department of Plant Physiology, Slovak University of Agriculture, Nitra, 949 01 Nitra, Slovakia; 10Department of Agronomy, Hajee Mohammad Danesh Science and Technology University, Dinajpur 5200, Bangladesh; shahid_sohana@yahoo.com; 11Department of Crop Science, Institute of Crop Science and Resource Conservation (INRES), University Bonn, 53115 Bonn, Germany; mhabibur@uni-bonn.de; 12Department of Agronomy, MNS-University of Agriculture-Multan-Pakistan, Punjab 66000, Pakistan; 13Department of Agronomy, Bahadur Sub-Campus Layyah, College of Agriculture, Bahauddin Zakariya University, Punjab 31200, Pakistan; wasayauaf@gmail.com; 14Department of Agronomy, Faculty of Agriculture, University of Poonch Rawalakot, Rawalakot 12350, Pakistan; aamir1801@yahoo.com; 15Department of Agronomy, Faculty of Agriculture, University of Kafrelsheikh, Kafr El-Sheikh 33516, Egypt

**Keywords:** drought, antioxidants, osmoprotectant, yield, selenium, camelina, canola

## Abstract

Drought poses a serious threat to oilseed crops by lowering yield and crop failures under prolonged spells. A multi-year field investigation was conducted to enhance the drought tolerance in four genotypes of Camelina and canola by selenium (Se) application. The principal aim of the research was to optimize the crop yield by eliciting the physio-biochemical attributes by alleviating the adverse effects of drought stress. Both crops were cultivated under control (normal irrigation) and drought stress (skipping irrigation at stages i.e., vegetative and reproductive) conditions. Four different treatments of Se viz., seed priming with Se (75 μM), foliar application of Se (7.06 μM), foliar application of Se + Seed priming with Se (7.06 μM and 75 μM, respectively) and control (without Se), were implemented at the vegetative and reproductive stages of both crops. Sodium selenite (Na_2_SeO_3_), an inorganic compound was used as Se sources for both seed priming and foliar application. Data regarding physiochemical, antioxidants, and yield components were recorded as response variables at crop maturity. Results indicated that WP, OP, TP, proline, TSS, TFAA, TPr, TS, total chlorophyll contents, osmoprotectant (GB, anthocyanin, TPC, and flavonoids), antioxidants (APX, SOD, POD, and CAT), and yield components (number of branches per plant, thousand seed weight, seed, and biological yields were significantly improved by foliar Se + priming Se in both crops under drought stress. Moreover, this treatment was also helpful in boosting yield attributes under irrigated (non-stress) conditions. Camelina genotypes responded better to Se application as seed priming and foliar spray than canola for both years. It has concluded that Se application (either foliar or priming) can potentially alleviate adverse effects of drought stress in camelina and canola by eliciting various physio-biochemicals attributes under drought stress. Furthermore, Se application was also helpful for crop health under irrigated condition.

## 1. Introduction

Water scarcity in changing climate scenarios has become a serious threat to sustainable crop production globally, and especially in developing countries of South Asia such as Pakistan [1]. Agricultural droughts pose a significant threat to crop failure leading to food shortages [2,3,4]. Water stress negatively affects the physiological and biochemical activities of the plant, including photosynthesis rate, osmotic potential, turgor pressure along with serious damages to the cellular membranes [5]. Agricultural yield can be maximized by evaluating and mitigating the impacts of drought on crops [6]. Numerous techniques have been used to protect against cellular and oxidative damages and to regulate enzymatic activities in crop plants under water stress conditions [7]. Selenium (Se) as basal or foliar fertilization is a current technology used to protect crop plants from drought stress. Selenite (IV, VI) are the major two species present in soil and can be used efficiently by plants. It has been reported that the application of Se increased proline accumulation but did not affect the water uptake capacity, nor did it have a negative impact on plant biomass under drought [8]. In foliar application, Se ions easily diffuse to epidermal cells as they are transported by xylem and phloem, hence becoming part of the plant body [9]. There are various studies carried out about the foliar application of Se on cereals [9], oilseed crops [8], carrots roots [10], onion and garlic bulbs [11,12], broccoli [13], etc., resulting in a significant improvement in the growth and yield of crops. Thus, Se application has the potential to increase the nutritional value of the crop, the grain yield as well as enable plants to have abiotic tolerance. Se improves plant growth by accumulating starch in the chloroplast [14]. Besides, Se can also regulate the activities of several antioxidant enzymes and metabolites providing oxidative stress tolerance to plants [15]. However, the toxicity or benefits of Se are highly dependent on the applied concentration [16,17]. Although, Se is not essential for plants as in the case of humans and animals, but numerous studies showed the beneficial effects of Se in plant health [18].

Furthermore, Se is a beneficial element that acts as an antioxidant and anti-senescent, it is also involved in active defense against abiotic and biotic stressors. It has been reported to be closely associated with higher antioxidant activity [19,20]. In plants, Se has chemical properties similar to sulfur (S) being present in sulfur amino acids such as selenomethionine (SeMet) and selenocysteine (SeCys). The Se is uptakes as selenite and transported via S-transporters leading to its transformation into SeCys and SeMet, which are usable forms in plants [21]. In an investigation, Klognerová et al. [22] reported that Se fortified oilseed rape plants have SeMet as a stable compound stored in tissues, while the same compound was predominately found in defatted oilseed rape meal [23]. The Se is involved in counteracting various types of abiotic stress experienced by plants such as cold, drought, heavy metal pollutants, and intense light [20,24,25]. However, higher concentrations (>1 ppm) can be toxic to plants, causing extreme physiological disturbances [25]. Exogenous application of Se can also help to uptake more plant nutrients as reported in wild plants [26], and various cereals [27,28].

Among oilseed crops, canola (*Brassica napus*) is held in special esteem worldwide as the second most important source of oil [29] with ~40–50% oil and 40% proteins (in rapeseed meal) which is of dietary importance for humans and animals [8]. Canola is considered as the third oilseed plant in the world after palm and soybean [30], and besides oil content, the rich chemistry of phenols, flavonoids, vitamins, and carotenoids make it to treat a good consumable oilseed crop [8]. The leaves of this plant are used as animal feed due to containing a higher concentration of protein and less fibrous [31]. Oil extracted from oilseed rape contains a very high concentration of C18 fatty acids (such as oleic acid and linoleic acid (C18:3, ~10% *v*/*v*)) making it a fatty acid-rich diet source [32].

Camelina (*Camelina sativa*) is also an oilseed crop cultivated in Pakistan, belonging to the Brassicaceae family. The camellia seed contains almost 35–45% oil, and the plants complete their life cycle within 60–90 days [33]. It is known that the growth and productivity of crops are reduced under drought stress due to reduced uptake of nutrients and physiological functions. Under such conditions, foliar application nutrients can improve plant growth and yield. Brassica species have been reported to accumulate Se affecting their physiology. Therefore, the present field research was carried out to study the effects of foliar-applied Se on water relations, physiochemical, osmoprotectant, and antioxidants activities of camelina and canola genotypes under drought stress condition.

## 2. Materials and Methods

### 2.1. Research Location, Duration and Plant Materials

This study was carried out in the 2016 and 2017 growing seasons in the field area of the Cholistan Institute of Desert Studies (CIDS), The Islamia University of Bahawalpur, Pakistan. Healthy, high-yielding seeds of camelina genotypes viz. Canadian camelina, and Australian camelina, and canola genotypes viz. UAF Canola (developed by the University of Agriculture Faisalabad), and AARI canola (developed by Ayub Agricultural Research Institute) were used as plant materials.

### 2.2. Treatments, Design and Experimentation

Four varieties of both oilseed (camelina and canola, two varieties each) were used in this experiment. The seeds were pre-sanitized with 3% H_2_O_2_ for 10 minutes, followed by vigorous washing with distilled water 10 times. Sodium selenite (Na_2_SeO_3_), an inorganic compound, was used as Se source, both for seed priming and foliar application. This salt is a colourless solid. The pentahydrate Na_2_SeO_3_(H_2_O)_5_ is the most common water-soluble selenium compound. For the seed priming with selenium, 5.1 mg L^−1^ Se solution was prepared in distilled water to be used as a stock solution for priming solution preparation. Four levels of Se were applied, T_1_ = Seed-priming (with 75 μM Se solution), T_2_ = foliar application of 7.06 μM Se solution, T_3_ = foliar Se + Seed priming (7.06 μM and 75 μM), and T_4_ = control receiving no Se supplementation. The healthy seeds were sown in the field following a randomized complete block design (RCBD). The research field was divided into two distinct patches representing normal and water-stressed conditions. Water stress was achieved by skipping irrigations in the vegetative and reproductive stages for both years in both oilseed crops. The irrigation requirement was calculated keeping in mind the daily requirement of the crop and the field capacity of the soil. Seed priming was done pre-sowing while foliar application of Se was done on 10-day-old seedlings for the whole growing season with the gap of 25 days.

### 2.3. Data Recording

#### 2.3.1. Water Potential Determination

A pressure chamber (Scholander type) was used to record water potential on the third fully matured leaf from the top tip of the plant. The leaves were collected and frozen at −20 °C following the method adopted by [34]. The osmotic potential was determined, and the difference method was used to get an idea of the water pressure potential inside the leaves of the plant as follows:(ψp) = (ψw) − (ψs)(1)
where (ψp), turgor potential; (ψs), osmotic potential, and (ψw), water potential.

#### 2.3.2. Biochemical Parameter Assay

##### Soluble Proteins, Amino Acids, Sugars, and Proline

The biochemical parameters under study, total soluble sugars (TSS), total soluble proteins (TSP), total free amino acid (TFAA) contents, and proline contents of leaves were measured following standard protocols [35,36,37,38,39].

##### Total Soluble Protein (TSP)

Total soluble proteins were determined using the procedures described by [35]:Reagents: For the preparation of phosphate buffer (0.2 M) solution, the following chemicals were used; a one-molar solution of NaH_2_PO_4_·2H_2_O (156.01 g L^−1^) was prepared as the stock and a one-molar solution of Di-sodium hydrogen phosphate (Na_2_HPO_4_·2H_2_O, 177.99 g L^−1^) was prepared as the stock.Copper Reagents: Solution A: Na_2_CO_3_ 2.0 g, NaOH 0.2 g and Sodium potassium tartrate 1.0 g. All three chemicals were dissolved in distilled water and the volume was made to 100 mL. Solution B: CuSO_4_·5H_2_O solution: 0.5 g CuSO_4_·5H_2_O was dissolved in 100 mL distilled water. Solution C: Fifty mL of solution A and 1.0 mL of solution B were mixed to prepare the alkaline solution. This solution was always prepared fresh.Folin phenol reagent: One hundred g of sodium tungstate and 25 g of sodium molybdate were dissolved in 700 mL of distilled water. Fifty mL of 85% orthophosphoric acid and 100 mL of HCl were added and the mixture was refluxed for 10 h. Then 150 g of lithium sulfate was added along with 50 mL of distilled water. A few drops of Br2 were also added.

The mixture was boiled without a condenser for 15 min to remove extra Br2. The mixture was then cooled and diluted to 1000 mL. Standard Bovine Serum Albumin (BSA) solution (1 µg mL^−1^). Ten mg of Bovine serum albumin (BSA) was dissolved in 10.0 mL of distilled water.

Extraction: Fresh leaf material (0.5 g) was chopped in 10 mL of phosphate buffer (0.2 M) of pH 7.0 and was ground. The ground leaf material was centrifuged at 5000 *g* for 5 min. The supernatant was used for protein determination.

Procedure: One mL of the leaf extract from each treatment was taken in a test tube. The blank contained 1 mL of phosphate buffer (pH 7.0). One mL of solution C was added to each test tube. The reagents in the test tube were thoroughly mixed and allowed to stand for 10 min at room temperature. Then 0.5 mL of Folin-Phenol reagent (1:1 diluted) was added, mixed well, and incubated for 30 min. at room temperature. The optical density (OD) was read at 620 nm on a spectrophotometer (Hitachi, 220, Tokyo, Japan).

##### Total Free Amino Acids (TFA)

Total free amino acids were determined according to [35,36,37]. Fresh plant leaves (0.5 g) were chopped and extracted with phosphate buffer (0.2 M) having pH 7.0. Took 1 mL of the extract in 25 mL test tube, added 1 mL of pyridine (10%) and 1 mL of ninhydrin (2%) solution in each test tube. Ninhydrin solution was prepared by dissolving 2 g ninhydrin in 100 mL distilled water. The test tubes with a sample mixture, heated in boiling water bath for about 30 min. The volume of each test tube was made up to 50 mL with distilled water. The optical density of the colored solution at 570 nm was recorded using a spectrophotometer. Developed a standard curve with Leucine and calculated free amino acids using the formulae given below:(2)$$\begin{array}{l} \mathrm{Total}\;\mathrm{amino}\;\mathrm{acids}\\ =\frac{\mathrm{Graph}\;\mathrm{reading}\;\mathrm{of}\;\mathrm{sample}\times \mathrm{Volume}\;\mathrm{of}\;\mathrm{factor}\times \mathrm{Dilution}\;\mathrm{factor}}{\mathrm{Weight}\;\mathrm{of}\;\mathrm{fresh}\;\mathrm{tissue}\times 1000} \end{array}$$

##### Total Soluble Sugars (TSS)

Total soluble sugars were determined according to the method of [38,39].

Reagents: Anthrone reagent was prepared by dissolving 150 mg of anthrone in 72% H_2_SO_4_ solution. This reagent was freshly prepared whenever needed.

Extraction: Dried plant material was ground well in a micromill and the material was sieved through a 1 mm sieve of micromill. Plant material (0.1 g) was extracted in 80% ethanol solution. The extract was incubated for 6 h at 60 °C. This extract was used for the estimation of total soluble sugars.

Procedure: Plant extract was taken in 25 mL test tubes and 6 mL anthrone reagent was added to each tube, heated in boiling water bath for 10 min. The test tubes were ice-cooled for 10 min. and incubated for 20 min. at room temperature (25 °C). Optical density was read at 625 nm on a spectrophotometer (Hitachi, 220, Tokyo, Japan). The concentration of soluble sugars was calculated from the standard curve developed by using the above method.

##### Proline Determination

The proline was determined according to the method of [39]. Fresh leaf material of 0.5 g was grounded and dissolved in 10 mL of 3% sulfosalicylic acid. The sample material was filtered by using Whatman No. 2 filter paper. Two mL of the filtrate was taken in a test tube and reacted with 2 mL acid ninhydrin solution. Acid ninhydrin solution was prepared by dissolving 1.25 g ninhydrine in 30 mL of glacial acetic acid and 20 mL of 6 M orthophosphoric acid.

Two mL of glacial acetic acid was added in the test tube and kept for 1 h at 100 °C. After terminating the reaction in an ice bath, the reaction mixture was extracted with 10 mL toluene. The continuous air stream was passed vigorously for 1–2 minutes in the reaction mixture. The chromophore containing toluene was aspirated from the aqueous phase, warmed at room temperature and the absorbance was noted at 520 nm on a spectrophotometer. Toluene was used as a blank. The proline concentration was calculated by using a standard curve and calculated on fresh weight basis as follows: Mmol proline g^−1^ fresh weight = (g proline mL^−1^ × mL of toluene/115.5)/(wt. of sample/5).

##### Osmoprotectants and Total Phenolic

For the determination of osmoprotectants and total phenolic contents, fresh plant leaves were grounded, and oven-dried followed by soaking in 10 mL methanol, water, and HCL solution (79: 20: 1 *v*/*v*), and the mixture was shaken for 72 h at 4 °C in dark. The above procedure extracted the essence of plant leaves, this mixture was centrifuged at 5000× *g* and its absorbance were measured at 530 and 657 nm following the protocol of Ribera-Fonseca et al. [40] using the following equations:Corrected A530 = A530 − 1/3 A657(3)
where A530 reading was obtained at 530 nm, similarly A657 obtained reading at 657 nm.

For the determination of Anthocyanins, and flavonoid content protocols of Zhao et al. [41] and Wang et al. [42] were used, respectively. For these, 0.5 mL extract volume was mixed with 0.5 mL of 2% AlCl_3_ solution in ethanol at room temperature. The absorbance of the solution mixture was calculated at 520 nm and flavonoid content determination was done making relation to equivalent Quercetin as a reference to the standard curve.

Glycine betaine (another osmoprotectant) was determination was done following the method proposed by Salama et al. [43]. For this purpose, 1 g fresh preserved plant leaves were grounded in 10 mL double deionized water and 1 mL mixture was spiked with 1 mL 2 N HCL for acidification of media. Half (0.5) mL of this acidified solution was taken in test tubes containing 0.2 mL of potassium triiodide (I3K), and the mixture was kept in an ice bath for 90 min. In this mixture, 1 mL of chilled and double deionized water and 1, 2 dichloroethane were introduced and steamed air was passed from the mixture for 1–2 min to completely dissolve solvents and break the double layer. The double beam spectrophotometer (Hitachi-150-20, Tokyo, Japan) was used for the determination of absorbance and ultimately GB contents were determined.

Soluble phenols determination was done following Mohd-Rosni et al. [36] method, 1 g leaf sample was grounded in 10 mL 80% acetone solution and the mixture was centrifuged at 4000 rpm of 10 min. Then, 20 μL of centrifuged material was loaded with 1.58 mL distilled water, 100 μL of Folin-Ciocalteu reagent, and 300 μL of sodium carbonate, and the mixture was kept at 40 °C in a hot bath for 30 min. Mixture absorbance was checked at 765 nm and a calibration curve was used for phenol concentration determination.

##### Assay of Antioxidant Enzyme Extraction

Antioxidant enzymes (APX, SOD, POD, and CAT) were determined following the Spectrophotometric technique as reported by [44]. The APX and SOD were determined using the methodology adopted by [45], CAT and POD were measured using the method described by [46].

##### Catalase (CAT)

Catalase activity was assayed by measuring the conversion rate of hydrogen peroxide to water and oxygen molecules. The activity was assayed in 3 mL reaction solution comprising 50 mM phosphate buffer with 7.0 pH, 5.9 mM of H_2_O_2_ and 0.1 mL enzyme extract. The catalase activity was determined by decline in absorbance at 240 nm after every 20 s due to consumption of H_2_O_2_. Absorbance change of 0.01 units min^−1^ was defined as one unit catalase activity.

##### Peroxidase (POX)

The activity of POD was determined by measuring the peroxidation of hydrogen peroxide with guaiacol as an electron donor [44]. The reaction solution for POD consists of 50 mM phosphate buffer with pH 5, 20 mM of guaiacol, 40 mM of H_2_O_2_ and 0.1 mL enzyme extract. The increase in the absorbance due to the formation of tetraguaiacol at 470 nm was assayed after every 20 s. One unit of the enzyme was considered as the amount of the enzyme that was responsible for the increase in OD value of 0.01 in 1 min. The enzyme activity was determined and expressed as unit’s min^−1^ g^−1^ fresh weight basis.

##### Ascorbate Peroxidase

Ascorbate peroxidase (APX) activity was measured by monitoring the decrease in absorbance of ascorbic acid at 290 nm (extinction coefficient 2.8 mM cm^−1^) in a 1 mL reaction mixture containing 50 mM phosphate buffer (pH 7.6), 0.1 mM Na-EDTA, 12 mM H_2_O_2_, 0.25 mM ascorbic acid and the sample extract as described by [44,45,46].

##### SPAD Value

A pre-calibrated handheld SPAD meter (SPAD-502) was used to measure the SPAD values.

##### Yield and Yield-Related Components

For biomass and growth parameter determination, the number of branched per plant, 100-grain weight, fresh and dry weights of tissue were recorded in tons per hectare.

### 2.4. Statistical Analysis

The recorded data were arranged and statistically analyzed using an ANOVA test (Table 1) at a 5% probability level followed by a pairwise comparison test using the least significant difference (LSD) test with the help of MSTAT-C software [47].

## 3. Results

### 3.1. Physiological Attributes

All the physiological attributes under study, such as water potential, osmotic potential, turgor pressure, and total chlorophyll contents (SPAD value of both oilseed crops were significantly reduced (*p* < 0.01) due to drought (Table 1). The reductions in all physiological attributes were observed where water deficit was applied at vegetative and reproductive stages. The graphical representation shows that both camelina genotypes performed better than both canola genotypes in terms of all physiological traits in both stages (vegetative and reproductive) with optimal water and limited water supply in both years (Figure 1).

Different treatments such as seed priming, foliar, and foliar plus seed priming with Se performed significantly improved all the physiological parameters of both oilseed crops with mentioned stages (vegetative and reproductive) in both years either with full and limited water supply, as shown in the graphical representation (Figure 1). All physiological characters showed a better improvement in camelina during the two stages (vegetative and reproductive) in both years compared to canola when Se was applied with seed priming plus the foliar application (T3) under limited water supply, as well as regular irrigation (Figure 1). However, the interactive effects were found to be non-significant for both years (Table 1).

### 3.2. Biochemical Attributes

All biochemical associations attributes such as total proline, total free amino acid, total soluble protein, and total soluble sugar were reduced significantly (*p* < 0.01) in the mentioned stages (vegetative and reproductive) for both varieties of camelina and canola in both years in the treatment with limited water application as presented in Table 1. The reduction in all biochemical attributes was noticed in limited water conditions (drought) water at both stages of canola and camelina in both years. The graphical representation shows that both genotypes of camelina performed better than canola genotypes in all biochemical characters under full application of water and limited water in both years (Figure 2). Different treatments such as seed priming, foliar, and foliar plus seed priming with Se performed significantly better by improving all the biochemical parameters of both crops in both stages and for both years either under full water provision and limited water conditions, as presented in Figure 2. In camelina, all the biochemical features evaluated showed better positive effects for both stages during both years compared to canola when foliar (T3) was applied with seed priming carried out under limited and regular irrigation (Figure 2). The interactions were not found significant as given in Table 1.

#### 3.2.1. Assay of Antioxidants

Different antioxidant attributes such as APX, SOD, POD, and CAT were significantly reduced (*p* < 0.01) in the mentioned stages (vegetative and reproductive) in both varieties of camelina and canola in both years in case of limited water application as presented in Table 1. The reduction in all antioxidant attributes was observed in the treatments with limited application of water, this response was not detected in the treatments with adequate water supply. All studied antioxidants, both genotypes of camelina performed better than both canola genotypes at both stages under partial and full application of water during both the years (Figure 3). Different treatments such as seed priming, foliar, and foliar plus seed priming with Se improved all the antioxidant parameters of both oilseed crops during both years either under full application of water and limited water conditions (Figure 3). All evaluated antioxidant elements showed better improvement in camelina in both critical stages during both years compared to canola when Se was applied as seed priming plus foliar application for the treatments carried out under limited and regular irrigation (Figure 3). The interactions were not found significant (Table 1).

#### 3.2.2. Osmoprotectants Characters

Osmoprotectant association attributes such as anthocyanin, flavonoid, GB, and TPC were significantly reduced (*p* < 0.01) in the vegetative and reproductive phases in both varieties of camelina and canola during both years with limited application of water (Table 1). The reduction in all osmoprotectant attributes was significantly greater with the limited application of water than with the full application of water in both canola and camelina stages during both years. Camelina genotypes performed better than the canola genotypes on all osmoprotectant traits in both stages under full and limited water application during both years (Figure 4). Different treatments such as seed priming, foliar and foliar plus seed priming with Se performed significantly well in improving all the osmoprotectant parameters of the crops in both stages during both years either with full and limited application of water (Figure 4). Camelina showed better improvements in all the osmoprotectant characters in both stages and years compared to canola when Se was applied with seed priming plus foliar application under limited as well as the full application of water (Figure 4). The interactions were not found significant as given in Table 1.

### 3.3. Yield Attributes

Yield attributes such as the number of branches per plant, thousand seed weight, seed and biological yields of both crops were significantly reduced (P < 0.01) due to limited water application during both years (Table 1). The data revealed that both camelina genotypes performed better than canola genotypes in terms of all the evaluated yield traits under full and limited water supply during both years (Figure 5). Different treatments such as seed priming, foliar and foliar together with the combined application of Se remained significantly effective in improving all yield parameters of both crops (Figure 5). Yield characters showed better improvement in camelina crop at both stages during both years than canola, in plants where Se was applied with seed priming plus foliar application under lower as well as the full application of water (Figure 5). The interactions were not found significant (Table 1).

## 4. Discussion

Both oilseed crops (camelina and canola) exhibited significant improvements in osmoregulation parameters like water-potential (Ψw), turgor-pressure, and osmotic potential under both control and water-limited conditions, when treated with Se either as seed priming or foliar application at both stages (vegetative and reproductive) tissues, the application of Se significantly enhanced the water uptake by the roots without decreasing the transpiration rate (E) [48]. The results of the present study revealed that the exogenous application of Se increased the Ψw and similar findings were reported by [49]. Likewise, a significant rise in biological and economical yield and water use efficiency (WUE) of the maize crop was seen by foliar-applied Se [50]. Similarly, it was observed that Se addition in plant growing media improved the leaf water potential of maize under severe water deficit conditions [51]. However, maize plant anthocyanin and flavonoids color pigments were moderately reduced under drought stress conditions. It is suggested that the desiccation of the leaves affects the synthesis of anthocyanin and flavonoids more than the Chl contents. In addition, the effect of Se on the pigments of corn leaves was observed for anthocyanin and flavonoids.

Anthocyanin is an important compound involved in the protection of cellular machinery against abiotic stress-mediated by oxidative stress [52]. Similarly, flavonoids are antioxidants that act as cellular protectors defending the plants against water stress [50]. Among plant biochemical machinery units, total chlorophyll, chl ‘a’, and chl ‘b’ are very important for optimal photosynthetic activity [53]. Drought (or partial water stress) can severely affect the production of chl content thus significantly decrease photosynthetic activity [54,55].

The results of the study revealed that a considerable reduction in total chlorophyll content was observed under drought stress. Previous studies showed that water deficit caused a reduction of Chl content in several oilseed crops such as sunflower [56] and other cereals including wheat [57], and corn [58]. Reactive oxygen species (ROS) effectively lessened the chlorophyll content of the leaves by damaging the chloroplasts [59]. The results of the present experiment confirmed the findings that Se seed treatment and foliar application resulted in higher chlorophyll content on the leaves of camelina and canola crops under normal and water-deficit conditions. The plants of the camelina and canola crops also sustained the maximum Chl contents due to Se supply at both stages (vegetative and reproductive). In contrast, drought stress significantly declined the Chl contents at later growth stages (vegetative and reproductive stage) in rapeseed plants reported by [60]. Seed priming with Se plus foliar application revealed the maximum increase in foliar Chl content in camelina and canola under both normal and limited water conditions.

The production of plant TSS as well as its involvement in drought stress management is well documented in soybean [61], wheat [62], and rice [61]. Our study also found similar results, reporting a net increase in TSS in camelina and canola upon exogenous application of Se (foliar as well as seed priming). In our work, plants have shown the maximum accumulation in the vegetative stage and a similar observation was made by [63]. Se supplementation to plants helps increase TSS accumulation. An investigation conducted by [49] also has similar observations for wheat crops and reports a significant increase in TSP and TFA upon exogenous application of Se. The exogenous application of Se has helped various crops like soybean [64] as well as trees like pear-jujube [63]. Increase TFA concentration by Se application may be the result of the modulation of amino acid metabolism that controls protein production and NO_3_ reductase activity [65]. The accumulation of TFA can help plants withstanding abiotic stress (such as osmotic stress) [66]. Similar results can be inferred from the present investigation added to the indirect beneficial effects of the application of Se.

The application of Se has shown a significant effect in the enhancement of proline accumulation in plant tissue under saline [67], cold [68], and metal stress [69]. In this research, the exogenous application of Se has shown a significant rush in proline contents of camelina and canola under drought. Higher proline contents are an indication of impaired protein biosynthesis [70] and its content is used to assess water stress due to its direct correlation with the extent of stress [56]. Our findings regarding Se mediated enhancement of proline content is in direct agreement with [51] who reported a 2–4 times increased proline concentration in wheat intending to improve tolerance to water stress.

Abiotic stress in the plant disturbs its biochemical machinery and produces dangerous ROS. To counteract this, plants tend to produce some antioxidants (SOD, POD, and CAT) [50] whose contents can be enhanced by applying exogenous amendments. The results of the present study showed an increase in antioxidants activity of camelina and canola plants under drought stress at both in the vegetative and reproductive stages. Previous studies in three canola cultivars indicated that foliar-applied Se has significantly altered enzyme activities to mitigate drought stress, supporting the present conclusion [71]. Among these parameters, the SOD concentration was significantly improved, indicating the plants response to drought stress by regulating the radical scavenging mechanism. Similarly, the application of Se to multiple canola varieties has been reported to trigger excessive amounts of SOD under drought stress, supporting the present findings [72]. Similar findings can be found in published works of [66] and [30] that have shown an increase in the antioxidant activity of wheat and barley under exogenous applied Se, which counteracts oxidative management. The maximum antioxidant activities were recorded in plants where Se was applied by foliar route with Se plus seed priming with Se at vegetative and reproductive stage of camelina and canola under water deficit conditions. An increase in antioxidants activities by exogenous Se supply under drought stress is in line with the findings of [51] who suggested that optimal Se supply is necessary to increase antioxidant activity in water deficit plants. As far as the mode of application is concerned, Se was not as effective as in a single-mode, rather it was helpful when applied synergistically as seed priming + foliar spray. The present conclusion was supported by Habibi [30], who depicted that foliar-applied Se alone is not helpful to combat the drought stress in Canola, indicated by higher accumulation of MDA content.

The application of Se influences the growth of plants but the cited results are not very consistent. It has a growth-promoting effect in soybean [64]. However, in the work conducted by [73], the authors reported a non-significant effect of Se application on soybean yield, similar results were reported for potato by Germ et al. [72]. In our work, exo-applied Se has shown a clear effect to enhance growth and yield of camelina and canola with foliar + seed priming. Between both crops, camelina responded better under water stress. The fact that camelina has better adaptability under drought stress compared to canola has been previously reported [74]. Hence, the role of camelina has become more proficient as an oilseed crop in the changing climate. In the present study, the application of Se as foliar + seed priming treatment (T3) resulted in higher grain yield compared to the control treatment. In previous studies carried out in two canola cultivars in the subtropical dryland zone where Se was applied at 1.5 mg/L and 3 mg/L, the results showed a significant contribution of this element to grain yield under drought stress [8]. This conclusion was also supported by the work published by Curtin et al. [75] which listed the foliar Se application method as more effective than fertilization for the wheat crop, as the foliar application of Se enhances its foliar diffusion making it more available to plants [76]. Exogenous application of Se is beneficial in general to tackle water stress in our study but a very high concentration should be avoided since it can damage the metabolism of the leaves and reduce the growth and plant production [19]. Therefore, it requires comprehensive further investigations.

## 5. Conclusions

Drought poses a substantial threat to oilseed crops including camelina and canola jeopardizing the food security of a rapidly growing population. However, supplementation of Se could alleviate the adverse effect of these unfavourable conditions through the alteration of the physio-biochemical process of plants. After two years of observation, it is revealed that when Se was used for seed priming and foliar application, water-related data (water potential, osmotic potential, turgor pressure), biochemical parameters (total soluble sugars, total free amino acids, total proteins, total sugars, total chl contents), osmoprotectants (glycine betaine, proline, anthocyanin, total phenolic contents, flavonoids) and antioxidants (ascorbate peroxidase, superoxide dismutase, peroxidase) and catalase improved significantly, ultimately leading to improving the yield and yield components of both crops under drought stress. Among these crops, camelina genotypes responded better to Se than canola during both years. Therefore, it can be concluded that the application of Se, either foliar or seed priming, can alleviate the adverse effect of drought stress in camelina and canola by eliciting various physio-biochemical attributes. Furthermore, oilseed crop varieties respond differently to Se seed priming and foliar application.

## Figures and Tables

**Figure 1 molecules-26-01699-f001:**
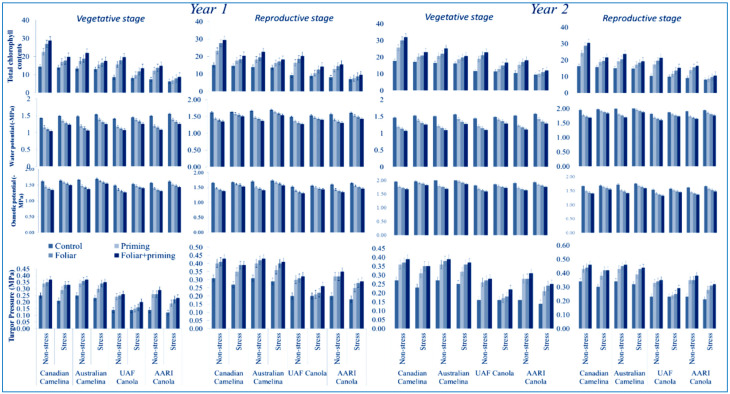
Impact of Se application (seed priming and foliar) on physiological attributes at both stages (vegetative and reproductive) of oilseed crops (camelina and canola) subjected to water deficit in the year 2016 and 2017 (mean values ± S.E).

**Figure 2 molecules-26-01699-f002:**
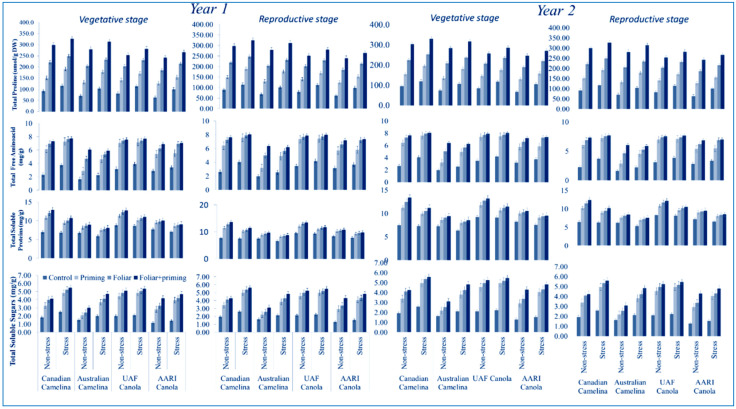
Impact of Se application (seed priming and foliar) on biochemical attributes at both stages (vegetative and reproductive) of oilseed crops (camelina and canola) subjected to water deficit in the year 2016 and 2017 (mean values ± S.E).

**Figure 3 molecules-26-01699-f003:**
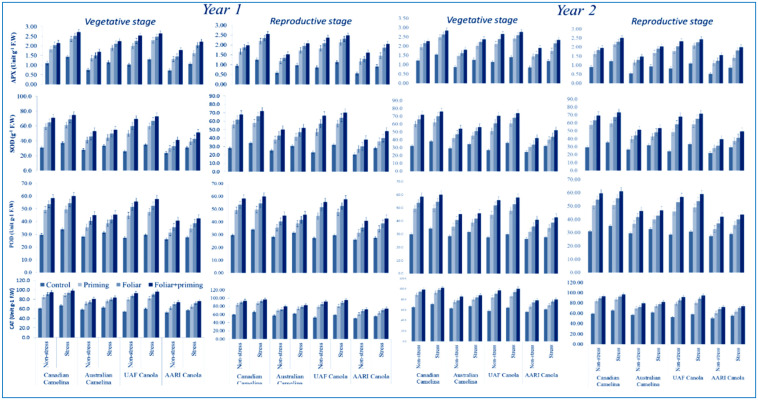
Impact of Se application (seed priming and foliar) on antioxidant activities at both stages (vegetative and reproductive) of oilseed crops (camelina and canola) subjected to water deficit in the year 2016–2017 (mean values ± S.E).

**Figure 4 molecules-26-01699-f004:**
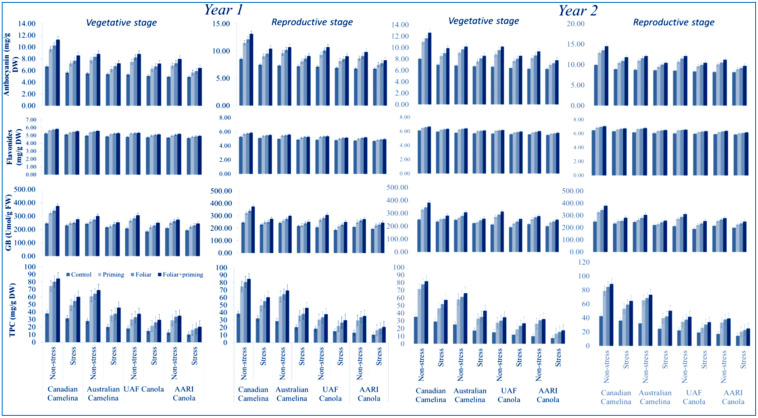
Impact of Se application (seed priming and foliar) on osmoprotectants at both stages (vegetative and reproductive) of oilseed crops (camelina and canola) subjected to water deficit in the year 2016–2017 (mean values ± S.E).

**Figure 5 molecules-26-01699-f005:**
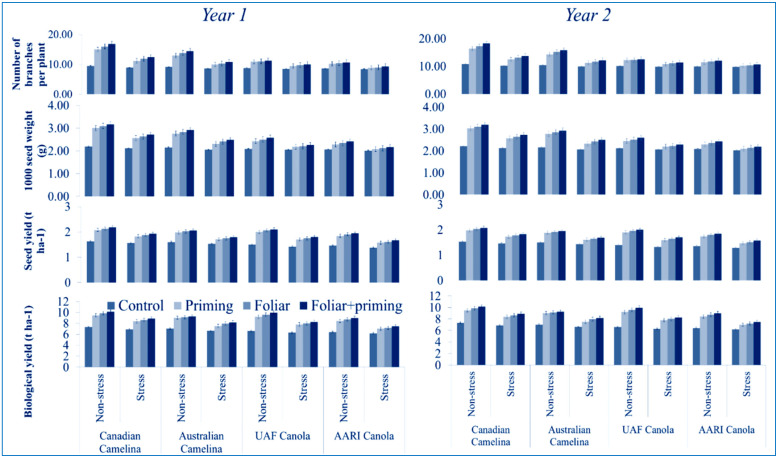
Impact of Se application (seed priming and foliar) on yield and yield components at both stages (vegetative and reproductive) of oilseed crops (camelina and canola) subjected to water deficit in the year 2016–2017 (mean values ± S.E).

**Table 1 molecules-26-01699-t001:** Analysis of variance (ANOVA) of growth, water relations, biochemical, osmoprotectants, antioxidant, and yield parameters of camelina and canola under drought stress by foliar-applied Selenium.

Traits	Treatments	Varieties	Treatment × Varieties
Total Chlorophyll Contents (SPAD)	***	**	**
Water Potential (−MPa)	**	**	**
Osmotic Potential (−MPa)	**	**	**
Turgor Pressure (MPa)	***	**	*
Total Proline (µmol/g DW)	***	**	NS
Total Soluble Sugars (µmol/g FW)	***	**	NS
Total Free Amino Acids (mg/g)	***	**	NS
Total Soluble Proteins (mg/g)	**	*	NS
Total Phenolic Contents (mg/g DW)	***	**	NS
Anthocyanin (mg/g DW)	***	**	NS
Flavonoids (mg/g DW)	***	**	NS
Glycine betaine (µmol/g FW)	***	**	NS
Ascorbic peroxidase (units’ min^−1^ g^−1^ FW)	***	**	NS
Superoxide Dismutase (units’ min^−1^ g^−1^ FW)	***	**	NS
Peroxidase (units’ min^−1^ g^−1^ FW)	***	**	NS
Catalase (units’ min^−1^ g^−1^ FW)	***	**	NS
Number of branches per plant	**	*	NS
Thousand seed weight (g)	**	*	NS
Seed yield (t/ha)	***	**	NS
Biological yield (t/ha)	**	**	NS

*, ** and *** = Significant at 0.05, 0.01 and 0.001 level, respectively; NS = Non-significant.

## Data Availability

Most of the recorded data are available in all Tables and Figures in the manuscript.

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
