# Peer review of "Selenium Alleviates the Adverse Effect of Drought in Oilseed Crops Camelina (Camelina sativa L.) and Canola (Brassica napus L.)"

_molecules, 2021, doi:10.3390/molecules26061699_

Round 1
Reviewer 1 Report
The Authors have significantly improved their work, however I still have not got the answer to my question what selenium compound was used to prepare solutions used in the study. There is also no satisfactory and clearly information in the Section 2.2 on the duration of the experiment. The reader is astonished to find in the Figures results gathered during TWO years of study.
There are also still some mistakes in the text.
Author Response
Response to Reviewer_1 comments
Comments and Suggestions for Authors
The Authors have significantly improved their work, however I still have not got the answer to my question what selenium compound was used to prepare solutions used in the study. There is also no satisfactory and clearly information in the Section 2.2 on the duration of the experiment. The reader is astonished to find in the Figures results gathered during TWO years of study.
Authors’ response: Thank you so much for your valuable remarks. Sodium selenite was used as Se sources for both seed priming and also foliar application. Sodium selenite is the inorganic compound with the formula Naâ‚‚SeO₃. This salt is a colourless solid. The pentahydrate Naâ‚‚SeO₃(Hâ‚‚O)â‚… is the most common water-soluble selenium compound. For seed priming with selenium, 5.1mg L-1 Se solution was prepared in distilled water to be used as a stock solution for priming solution preparation. Four levels of Se were applied, T1 = Seed-priming (with 75 μM Se solution), T2 = foliar application of 7.06 μM Se solution, T3 = foliar Se + Seed priming (7.06 μM and 75μM), and T4 = control receiving no Se supplementation. Seed priming was done pre-sowing while foliar application of Se was done on seedlings of 10 day’s age for the whole growing season with the gap of 25 days.
There are also still some mistakes in the text.
Authors’ response: We have thoroughly checked the whole manuscript and have fixed all errors. Please check all edits in red marks in the text of the manuscript.

Reviewer 2 Report
- The manuscript is devoted to the study of the properties of selenium in plants under drought conditions. The problem of growing plants in conditions of recurring drought is extremely urgent. The authors propose to use selenium to increase plant resistance to drought. Selenium is considered an element that increases plant stress resistance to various factors. I propose to write about this in the introduction to the manuscript. For example, it is known that selenium makes it easier to tolerate salinity (10.3390 / ijms21010148), hyperthermia (10.1021 / acsomega.0c02448) and even phytopathogens (10.1007 / s00253-019-10334-y).
- The authors use a very unusual manner of coloring the text in the manuscript. I suggest making all letters and numbers in the text black.
- From the section of the manuscript materials and methods it is not at all clear what chemical compound of selenium the authors use. What is its purity? Origin? Is it solution or colloid? This information is necessary to understand the processes described in the manuscript.
- Why is the third leaf of the plant taken for hydrodynamic studies? Why not the second or the first? Or the fourth?
- 2.3.2.1. Soluble Proteins, Amino acids, Sugars, and Proline. - a detailed description of the applied techniques is required.
- 2.3.2.3. Assay of Antioxidant Enzyme Extraction. - a detailed description of the applied techniques is required.
- SPAD decryption is required. How was chlorophyll concentration measured? Commercial appliance? Own installation?
- The recorded data were arranged and statistically analyzed by employing ANOVA at 5% probability level [47] followed by a pairwise comparison was made by using least significant difference (LSD) test with the help of MSTAT-C software. What does reference 47 have to do with statistics? The link refers to measurements and should be at the end of the sentence. Moreover, the link is not correct. In a real article, other authors ... This needs to be fixed.
- Table 1 remains very difficult to understand. The table shows the levels of significance of differences, but it is not written from what all these indicators differ. This needs to be fixed!
- After viewing Figure 1, the question appears. Does stress affect the studied parameters of plants? apparently not in most cases ... What is the point in the study if not stressed and stressed plants do not differ from each other in most indicators? If these differences exist, then it probably makes sense to show them!
On the whole, the manuscript needs substantial revision. There are many points that need clarification, without which reading the manuscript does not make as much sense as we would like. In this form, I cannot recommend the manuscript for publication.
Author Response
Response to Reviewer_2 comments
Comments and Suggestions for Authors
- The manuscript is devoted to the study of the properties of selenium in plants under drought conditions. The problem of growing plants in conditions of recurring drought is extremely urgent. The authors propose to use selenium to increase plant resistance to drought. Selenium is considered an element that increases plant stress resistance to various factors. I propose to write about this in the introduction to the manuscript. For example, it is known that selenium makes it easier to tolerate salinity (10.3390 /ijms21010148), hyperthermia (10.1021/acsomega.0c02448) and even phytopathogens (10.1007/s00253-019-10334-y).
Authors’ response: Many thanks for your informative suggestion, which have allowed us to considerable improvement of the manuscript. Please check all edits as red marks in the text of the manuscript.
- The authors use a very unusual manner of coloring the text in the manuscript. I suggest making all letters and numbers in the text black.
Authors’ response: We are fully agreed with your suggestion, however, we have already edited the whole manuscript and showing in red marks.
- From the section of the manuscript materials and methods it is not at all clear what chemical compound of selenium the authors use. What is its purity? Origin? Is it solution or colloid? This information is necessary to understand the processes described in the manuscript.
Authors’ response: Thank you so much for your valuable remarks. Sodium selenite was used as Se sources for both seed priming and also foliar application. Sodium selenite is the inorganic compound with the formula Naâ‚‚SeO₃. This salt is a colourless solid. The pentahydrate Naâ‚‚SeO₃(Hâ‚‚O)â‚… is the most common water-soluble selenium compound. For seed priming with selenium, 5.1mg L-1 Se solution was prepared in distilled water to be used as a stock solution for priming solution preparation. Four levels of Se were applied, T1 = Seed-priming (with 75 μM Se solution), T2 = foliar application of 7.06 μM Se solution, T3 = foliar Se + Seed priming (7.06 μM and 75μM), and T4 = control receiving no Se supplementation. Seed priming was done pre-sowing while foliar application of Se was done on seedlings of 10 day’s age for the whole growing season with the gap of 25 days.
- Why is the third leaf of the plant taken for hydrodynamic studies? Why not the second or the first? Or the fourth?
Authors’ response: Measuring the hydrodynamic studies of plants is tricky and just for general way we take leaf.
- 2.3.2.1. Soluble Proteins, Amino acids, Sugars, and Proline. - a detailed description of the applied techniques is required.
Authors’ response: Details biochemical procedures of Soluble Proteins, Amino acids, Sugars, and Proline are now added in the suggestion sections
- 2.3.2.3. Assay of Antioxidant Enzyme Extraction. - a detailed description of the applied techniques is required.
Authors’ response: Details biochemical procedures of Assay of Antioxidant Enzyme Extraction is now added in the suggestion sections
- SPAD decryption is required. How was chlorophyll concentration measured? Commercial appliance? Own installation?
Authors’ response: SPAD value is the measurement of chlorophyll content
- The recorded data were arranged and statistically analyzed by employing ANOVA at 5% probability level [47] followed by a pairwise comparison was made by using least significant difference (LSD) test with the help of MSTAT-C software. What does reference 47 have to do with statistics? The link refers to measurements and should be at the end of the sentence. Moreover, the link is not correct. In a real article, other authors ... This needs to be fixed.
Authors’ response: We have replaced the citation at the end of the sentence
- Table 1 remains very difficult to understand. The table shows the levels of significance of differences, but it is not written from what all these indicators differ. This needs to be fixed!
Authors’ response: The suggestion has been made
- After viewing Figure 1, the question appears. Does stress affect the studied parameters of plants? apparently not in most cases ... What is the point in the study if not stressed and stressed plants do not differ from each other in most indicators? If these differences exist, then it probably makes sense to show them!
Authors’ response: All figures have been improved to understand clearly
On the whole, the manuscript needs substantial revision. There are many points that need clarification, without which reading the manuscript does not make as much sense as we would like. In this form, I cannot recommend the manuscript for publication.
Authors’ response: Many thanks for your informative suggestion, which have allowed us to considerable improvement of the manuscript. We have thoroughly checked the whole manuscript and have edited where necessary. Please check all edits as red marks in the text of the manuscript.
Reviewer 3 Report
The manuscript entitled ''Selenium Alleviated the Adverse Effect of Drought in Camelina (Camelina sativa L.) and Canola (Brassica napus L.) Through Physio-Biochemical Alterations'', studies the effects of drought stress on Camelina and Canola, as well as the abilities of exogenous Selenium for mitigate impact of adverse condition. Thus, application of exogenous Selenium can be recommended as primary objective to increase the crop yield in drought stress. However, the quality of the MS is below standard and needs rigorous improvement before it is fit for publication. The text is hard to read with an unclear structure.
The authors need to revise the title of the paper in a more meaningful way. It is too long. Please shorten it.
The abstract is written in a way lacks logic. It should highlight the salient findings more critically.
Introduction:
lines 60-62; delete, “Different types of droughts like hydrological, meteorological, and agricultural cause precipitation deficiency in certain regions having hotter climate for most part of the year”.
Lines 66-68: “Numerous techniques have been used to protect against cellular and oxidative damage and regulate enzymatic activities under water stress conditions in crop plants.”, cite examples of technologies used for this purpose. I suggest the review and quote: Filgueiras, L.; Silva, R.; Almeida, I.; Vidal, M.; Baldani, JI; Meneses, CHSG. Gluconacetobacter diazotrophicus mitigates drought stress in Oryza sativa. Plant Soil 451, 57–73 (2020). https://doi.org/10.1007/s11104-019-04163-1
Line 87: insert; “In plants, Se has chemical properties similar to sulfur, being present in sulfur amino acids such as selenomethionine and selenocysteine.
Materials and Methods:
Lines 132-134: Why were these concentrations used? Were any references removed? or was a previous experiment carried out?
Lines 152-154: detail how the techniques were performed. The quantification of free proline has to be in topic 2.3.2.2. Osmoprotectants and total phenolic.
Lines 187-190: detail how the techniques were performed.
Lines 192-193; 204: Determine a correlation equation between the SPAD method and the organic extraction of chlorophyll, so the authors can talk about total chlorophyll contents.
Statistical Analysis: Authors should discuss the results integrally. The discussion is based on individual results. I suggest that integrating the results will give more value to the work. The response of the plant to the drought is integrated. I suggest that you discuss by integrating all your results. You can use correlation tests (PCA or Pearson Correlation). The authors reported in the Materials and Methods that they used the LSD Test to differentiate means, but suppressed the test results in all the graphs, please insert all the letter distributions of the test.
Results: The results of this study are not fully explained therefore the interpretation of the results is very difficult. The author needs to provide the % increase or decrease rather than just writing ''significantly increased….''.
Discussion:
Line 388: @ ?????
The discussion is poorly written hence, needs rewriting. The discussion should be further strengthened by adding some more relevant papers. The literature search is insufficient, only few related research papers in the past three years are cited, add the latest research results appropriately. The authors report that there was a mitigation of adverse condition by applying of Selenium. Compare with the action of other factors found in plants under mitigation process. I suggest the review and quote: Silva, R.; Filgueiras, L.; Santos, B.; Coelho, M.; Silva, M.; Estrada-Bonilla, G.; Vidal, M.; Baldani, J.I.; Meneses, C. Gluconacetobacter diazotrophicus Changes The Molecular Mechanisms of Root Development in Oryza sativa Growing Under Water Stress. Int. J. Mol. Sci. 2020, 21, 333. https://doi.org/10.3390/ijms21010333
Rewrite the conclusion! It needs to be much improved.
Author Response
Response to Reviewer_3 comments
Comments and Suggestions for Authors
The manuscript entitled ''Selenium Alleviated the Adverse Effect of Drought in Camelina (Camelina sativa L.) and Canola (Brassica napus L.) through Physio-Biochemical Alterations'', studies the effects of drought stress on Camelina and Canola, as well as the abilities of exogenous Selenium for mitigate impact of adverse condition. Thus, application of exogenous Selenium can be recommended as primary objective to increase the crop yield in drought stress. However, the quality of the MS is below standard and needs rigorous improvement before it is fit for publication. The text is hard to read with an unclear structure.
Authors’ response: Thanks for your good comments and suggestions. We have thoroughly checked the whole manuscript and have fixed all errors. Please check all edits in red marks in the text of the manuscript.
The authors need to revise the title of the paper in a more meaningful way. It is too long. Please shorten it.
Authors’ response: As per your suggestion, we have been revised the title as: ''Selenium Alleviates the Adverse Effect of Drought in Oil Seed Crops Camelina (Camelina sativa L.) and Canola (Brassica napus L.)''
The abstract is written in a way lacks logic. It should highlight the salient findings more critically.
Authors’ response: We have rewritten the Abstract. Please check all edits in red marks in the text of the Abstract.
Introduction:
lines 60-62; delete, “Different types of droughts like hydrological, meteorological, and agricultural cause precipitation deficiency in certain regions having hotter climate for most part of the year”.
Authors’ response: Deleted
Lines 66-68: “Numerous techniques have been used to protect against cellular and oxidative damage and regulate enzymatic activities under water stress conditions in crop plants.”, cite examples of technologies used for this purpose. I suggest the review and quote: Filgueiras, L.; Silva, R.; Almeida, I.; Vidal, M.; Baldani, JI; Meneses, CHSG. Gluconacetobacter diazotrophicus mitigates drought stress in Oryza sativa. Plant Soil 451, 57–73 (2020). https://doi.org/10.1007/s11104-019-04163-1
Authors’ response: Added the citation
Line 87: insert; “In plants, Se has chemical properties similar to sulfur, being present in sulfur amino acids such as selenomethionine and selenocysteine.
Authors’ response: The suggested edit has been done. Please check the edit in red marks in the text of the manuscript.
Materials and Methods:
Lines 132-134: Why were these concentrations used? Were any references removed? or was a previous experiment carried out?
Authors’ response: Yes, several earlier findings have been used these amount of Se concentration for numerous field crops under abiotic stresses
Lines 152-154: detail how the techniques were performed. The quantification of free proline has to be in topic 2.3.2.2. Osmoprotectants and total phenolic.
Authors’ response: For meaningful this sub-section, we have been revised this section.
Lines 187-190: detail how the techniques were performed.
Authors’ response: The suggestion has been incorporated in the specific sentences
Lines 192-193; 204: Determine a correlation equation between the SPAD method and the organic extraction of chlorophyll, so the authors can talk about total chlorophyll contents.
Authors’ response: Chlorophyll contents for different treatments have been discussed in results sections. Please check all edits in red marks.
Statistical Analysis: Authors should discuss the results integrally. The discussion is based on individual results. I suggest that integrating the results will give more value to the work. The response of the plant to the drought is integrated. I suggest that you discuss by integrating all your results. You can use correlation tests (PCA or Pearson Correlation). The authors reported in the Materials and Methods that they used the LSD Test to differentiate means, but suppressed the test results in all the graphs, please insert all the letter distributions of the test.
Authors’ response: In the present study, the recorded data were arranged and statistically analyzed by employing ANOVA (Table 1) at a 5% probability level [47] followed by a pairwise comparison test by using the least significant difference (LSD) test with the help of MSTAT-C software.
Results: The results of this study are not fully explained therefore the interpretation of the results is very difficult. The author needs to provide the % increase or decrease rather than just writing ''significantly increased….''.
Authors’ response: Thank you so much for your good suggestion. We have revised the results section.
Discussion:
Line 388: @ ?????
Authors’ response: @ has been removed by ‘at’
The discussion is poorly written hence, needs rewriting. The discussion should be further strengthened by adding some more relevant papers. The literature search is insufficient, only few related research papers in the past three years are cited, add the latest research results appropriately. The authors report that there was a mitigation of adverse condition by applying of Selenium. Compare with the action of other factors found in plants under mitigation process. I suggest the review and quote:
Silva, R.; Filgueiras, L.; Santos, B.; Coelho, M.; Silva, M.; Estrada-Bonilla, G.; Vidal, M.; Baldani, J.I.; Meneses, C. Gluconacetobacter diazotrophicus Changes The Molecular Mechanisms of Root Development in Oryza sativa Growing Under Water Stress. Int. J. Mol. Sci. 2020, 21, 333. https://doi.org/10.3390/ijms21010333
Authors’ response: Thank you so much for your good suggestion. We have been edited the discussion section by adding latest citations including the mentioned one also.
Rewrite the conclusion! It needs to be much improved.
Authors’ response: Conclusion has been rewritten
Round 2
Reviewer 2 Report
The reviewer did not receive answers to most of the questions posed, or, in several cases, reviewer could not read a number of answers. The reviewer believes that the answers to the questions should be detailed.
Author Response
Response to Reviewer_2 comments
Dear Sir,
Thanks very much for your efforts and useful comments about our manuscript. We are very grateful and appreciate your good comments that help us to make the paper more quality and accurate. We responded to all the comments of the reviewers and heighted the changes in the manuscript, so anyone can follow the corrections. We have modified the manuscript accordingly, and the detailed corrections are listed below point by point and we hope this edition is more suitable and cover all the comments.
Comments and Suggestions for Authors
- The manuscript is devoted to the study of the properties of selenium in plants under drought conditions. The problem of growing plants in conditions of recurring drought is extremely urgent. The authors propose to use selenium to increase plant resistance to drought. Selenium is considered an element that increases plant stress resistance to various factors. I propose to write about this in the introduction to the manuscript. For example, it is known that selenium makes it easier to tolerate salinity (10.3390 /ijms21010148), hyperthermia (10.1021/acsomega.0c02448) and even phytopathogens (10.1007/s00253-019-10334-y).
Authors’ response: Many thanks for your informative suggestion, which have allowed us to considerable improvement of the manuscript. Please check all edits as red marks in the text of the manuscript.
- The authors use a very unusual manner of coloring the text in the manuscript. I suggest making all letters and numbers in the text black.
Authors’ response: We are fully agreed with your suggestion, however, we have already edited the whole manuscript and showing in red marks.
- From the section of the manuscript materials and methods it is not at all clear what chemical compound of selenium the authors use. What is its purity? Origin? Is it solution or colloid? This information is necessary to understand the processes described in the manuscript.
Authors’ response: Thank you so much for your valuable remarks. Sodium selenite was used as Se sources for both seed priming and also foliar application. Sodium selenite is the inorganic compound with the formula Naâ‚‚SeO₃. This salt is a colourless solid. The pentahydrate Naâ‚‚SeO₃(Hâ‚‚O)â‚… is the most common water-soluble selenium compound. For seed priming with selenium, 5.1mg L-1 Se solution was prepared in distilled water to be used as a stock solution for priming solution preparation. Four levels of Se were applied, T1 = Seed-priming (with 75 μM Se solution), T2 = foliar application of 7.06 μM Se solution, T3 = foliar Se + Seed priming (7.06 μM and 75μM), and T4 = control receiving no Se supplementation. Seed priming was done pre-sowing while foliar application of Se was done on seedlings of 10 day’s age for the whole growing season with the gap of 25 days.
- Why is the third leaf of the plant taken for hydrodynamic studies? Why not the second or the first? Or the fourth?
Authors’ response: We select fully expand young leaf for measurement, That’s way wo select 3rd leaf because the 1st and 2nd leaf are not fully expand and 4th leaf are old as compared to 3rd young leaf.
- 2.3.2.1. Soluble Proteins, Amino acids, Sugars, and Proline. - a detailed description of the applied techniques is required.
Authors’ response: Details biochemical procedures of Soluble Proteins, Amino acids, Sugars, and Proline are now added in the suggestion sections
2.3.2.1.1. Total soluble protein (TSP)
Total soluble proteins were determined using the procedures described by [35]:
Reagents: For the preparation of phosphate buffer (0.2 M) solution, the following chemicals were used; a one-molar solution of NaH2PO4.2H2O (156.01 g L-1) was prepared as the stock and a one-molar solution of Di-sodium hydrogen phosphate (Na2HPO4.2H2O, 177.99 g L-1) was prepared as the stock.
Copper Reagents: Solution A: Na2CO3 2.0 g, NaOH 0.2 g and Sodium potassium tartrate 1.0 g. All three chemicals were dissolved in distilled water and the volume was made to 100 mL. Solution B: CuSO4.5H2O solution: 0.5g CuSO4.5H2O was dissolved in 100 mL distilled water. Solution C: Fifty mL of solution A and 1.0 mL of solution B were mixed to prepare the alkaline solution. This solution was always prepared fresh.
Folin phenol reagent: One hundred g of sodium tungstate and 25 g of sodium molybdate were dissolved in 700 mL of distilled water. Fifty mL of 85% orthophosphoric acid and 100 mL of HCl were added and the mixture was refluxed for 10 h. Then 150 g of lithium sulfate was added along with 50 mL of distilled water. A few drops of Br2 were also added.
The mixture was boiled without a condenser for 15 min to remove extra Br2. The mixture was then cooled and diluted to 1000 mL. Standard Bovine Serum Albumin (BSA) solution (1 µg mL-1). Ten mg of Bovine serum albumin (BSA) was dissolved in 10.0 mL of distilled water.
Extraction: Fresh leaf material (0.5 g) was chopped in 10 mL of phosphate buffer (0.2 M) of pH 7.0 and was ground. The ground leaf material was centrifuged at 5000 g for 5 min. The supernatant was used for protein determination.
Procedure: One mL of the leaf extract from each treatment was taken in a test tube. The blank contained 1 mL of phosphate buffer (pH 7.0). One mL of solution C was added to each test tube. The reagents in the test tube were thoroughly mixed and allowed to stand for 10 min at room temperature. Then 0.5 mL of Folin-Phenol reagent (1:1 diluted) was added, mixed well and incubated for 30 min. at room temperature. The optical density (OD) was read at 620 nm on a spectrophotometer (Hitachi, 220, Japan).
2.3.2.1.2. Total free amino acids (TFA)
Total free amino acids were determined according to [35-37]. Fresh plant leaves (0.5 g) were chopped and extracted with phosphate buffer (0.2 M) having pH 7.0. Took 1 mL of the extract in 25 mL test tube, added 1 mL of pyridine (10%) and 1mL of ninhydrin (2%) solution in each test tube. Ninhydrin solution was prepared by dissolving 2 g ninhydrin in 100 mL distilled water. The test tubes with a sample mixture, heated in boiling water bath for about 30 min. Volume of each test tube was made up to 50 mL with distilled water. Read the optical density of the coloured solution at 570 nm using spectrophotometer. Developed a standard curve with Leucine and calculated free amino acids using the formulae given below:
Graph reading of sample x Volume of factor x Dilution factor
Total amino acids = ------------------------------------------------------------------------------
Weight of fresh tissue x 1000
2.3.2.1.3. Total soluble sugars (TSS)
Total soluble sugars were determined according to the method of [38,39].
Reagents: Anthrone reagent was prepared by dissolving 150 mg of anthrone in 72% H2SO4 solution. This reagent was freshly prepared whenever needed.
Extraction: Dried plant material was ground well in a micromill and the material was sieved through a 1 mm sieve of micromill. Plant material (0.1 g) was extracted in 80% ethanol solution. The extract was incubated for 6 h at 60oC. This extract was used for the estimation of total soluble sugars.
Procedure: Plant extract was taken in 25 mL test tubes and 6 mL anthrone reagent was added to each tube, heated in boiling water bath for 10 min. The test tubes were ice-cooled for 10 min. and incubated for 20 min. at room temperature (25oC). Optical density was read at 625 nm on a spectrophotometer (Hitatchi, 220, Japan). The concentration of soluble sugars was calculated from the standard curve developed by using the above method.
2.3.2.1.4. Proline determination
The proline was determined according to the method of [39]. Fresh leaf material of 0.5 g was ground and dissolved in 10 mL of 3% sulfo-salicylic acid. The sample material was filtered by using Whatman No. 2 filter paper. Two mL of the filterate was taken in a test tube and reacted with 2 mL acid ninhydrin solution. Acid ninhydrin solution was prepared by dissolving 1.25 g ninhydrine in 30 mL of glacial acetic acid and 20 mL of 6 M orthophosphoric acid.
Two mL of glacial acetic acid was added in the test tube and kept for 1 h at 100oC. After terminating the reaction in an ice bath, the reaction mixture was extracted with 10 mL toluene. Continuous air stream was passed vigorously for 1-2 minutes in the reaction mixture. The chromophore containing toluene was aspirated from the aqueous phase, warmed at room temperature and the absorbance was noted at 520 nm on spectrophotometer. Toluene was used as a blank. The proline concentration was calculated by using a standard curve and calculated on fresh weight basis as follows: Mmol proline g-1 fresh weight = (g proline mL-1 x mL of toluene/115.5) / (wt. of sample/5)
- 2.3.2.3. Assay of Antioxidant Enzyme Extraction. - a detailed description of the applied techniques is required.
Authors’ response: Details biochemical procedures of Assay of Antioxidant Enzyme Extraction is now added in the suggestion sections
2.3.2.3.1. Catalase (CAT)
Catalase activity was assayed by measuring the conversion rate of hydrogen peroxide to water and oxygen molecules, following the method described by [46]. The activity was assayed in 3 mL reaction solution comprising 50 mM phosphate buffer with 7.0 pH, 5.9 mM of H2O2 and 0.1 mL enzyme extract. The catalase activity was determined by decline in absorbance at 240 nm after every 20 sec due to consumption of H2O2. Absorbance change of 0.01 units min-1 was defined as one unit catalase activity.
2.3.2.3.2.Peroxidase (POX)
The activity of POD was determined by measuring peroxidation of hydrogen peroxide with guaiacol as an electron donor [44]. The reaction solution for POD consists of 50 mM phosphate buffer with pH 5, 20 mM of guaiacol, 40 mM of H2O2 and 0.1 mL enzyme extract. The increase in the absorbance due to the formation of tetraguaiacol at 470 nm was assayed after every 20 sec. One unit of the enzyme was considered as the amount of the enzyme that was responsible for the increase in OD value of 0.01 in 1 min. The enzyme activity was determined and expressed as units min-1 g-1 fresh weight basis.
2.3.2.3.3. Ascorbate peroxidase activity
Ascorbate peroxidase (APX) activity was measured by monitoring the decrease in absorbance of ascorbic acid at 290 nm (extinction coefficient 2.8 mM cm-1) in a 1 ml reaction mixture containing 50 mM phosphate buffer (pH 7.6), 0.1 mM Na-EDTA, 12 mM H2O2, 0.25 mM ascorbic acid and the sample extract as described by [44-46].
- SPAD decryption is required. How was chlorophyll concentration measured? Commercial appliance? Own installation?
Authors’ response: SPAD is the instrument that used for the measurement of total SPAD value. That’s value is the measurement of total chlorophyll contents
- The recorded data were arranged and statistically analyzed by employing ANOVA at 5% probability level [47] followed by a pairwise comparison was made by using least significant difference (LSD) test with the help of MSTAT-C software. What does reference 47 have to do with statistics? The link refers to measurements and should be at the end of the sentence. Moreover, the link is not correct. In a real article, other authors ... This needs to be fixed.
Authors’ response: We have replaced the citation at the end of the sentence
The recorded data were arranged and statistically analyzed using an ANOVA test (Table 1) at a 5% probability level followed by a pairwise comparison test using the least significant difference (LSD) test with the help of MSTAT-C software [47].
- Table 1 remains very difficult to understand. The table shows the levels of significance of differences, but it is not written from what all these indicators differ. This needs to be fixed!
Authors’ response: The suggestion has been made
*, **, *** = Significant at 0.05, 0.01 and 0.001 level respectively; NS = Non-significant.
- After viewing Figure 1, the question appears. Does stress affect the studied parameters of plants? apparently not in most cases ... What is the point in the study if not stressed and stressed plants do not differ from each other in most indicators? If these differences exist, then it probably makes sense to show them!
Authors’ response: All figures have been improved to understand clearly
On the whole, the manuscript needs substantial revision. There are many points that need clarification, without which reading the manuscript does not make as much sense as we would like. In this form, I cannot recommend the manuscript for publication.
Authors’ response: Many thanks for your informative suggestion, which have allowed us to considerable improvement of the manuscript. We have thoroughly checked the whole manuscript and have edited where necessary. Please check all edits as red marks in the text of the manuscript.

Reviewer 3 Report
Thanks for attending all the suggestions. The manuscript has been significantly improved. In view of the above, I believe that the article presents robust and consolidated content, bringing to light new information on changes Induced by selenium in camelina and canola under water deficit. I consider that the work has enough quality to be considered for publication in Molecules (MDPI).
Author Response
Comments and Suggestions for Authors
Thanks for attending all the suggestions. The manuscript has been significantly improved. In view of the above, I believe that the article presents robust and consolidated content, bringing to light new information on changes Induced by selenium in camelina and canola under water deficit. I consider that the work has enough quality to be considered for publication in Molecules (MDPI).
Authors’ response: Thank you so much for your valuable remarks

This manuscript is a resubmission of an earlier submission. The following is a list of the peer review reports and author responses from that submission.
Round 1
Reviewer 1 Report
The presented manuscript deals with the very important issue of the methods of alleviation of the destructive effects of drought on crops. The work contains the results of long lasting investigations on the influence of selenium on the yield and biochemical characteristics of oilseed plants cultivated in normal and water deficient conditions. The scope of gathered data is really impressive, but its presentation and discussion is very poor.
The manuscript would need very thorough revision:
- The paper is written very carelessly. There are a lot of missing spaces between words (for example: l. 99 Brassicanapus), words written incorrectly with capital letter (for example: l. 104 Oleic, l. 206 Carmelina), missing letters in words, strange symbols (for example: l. 41 @), omitted superscripts (for example: l.153 0oC), repeated words (for instance: l.147-148 following protocols given protocols)
- Language must be improved. There are cases of bad spelling and grammatical mistakes. Some sentences are difficult to understand (for example: lines 73-74, 110-113, 230-232,241-243, 332-333 and others). Their sense is blurred by grammatical and spelling mistakes or pure negligence.
- There are factual errors (for example: l. 104 two different formulas of linoleic acid)
- The objects of the study, oilseed plants, are described unclearly. In the introduction we have Brassica napus (l.97-105) and camelina (l.106-107). The name canola appears for the first time in line 120. Canadian camelina appears only in Figures. The abbreviation AARI (l.120-121) is not explained. I understand that AARI is the institution, so in this case what does „AARI canola” mean? Is it a special variety bred in this institute or just the variety which seeds were obtained from there?
There should be a clear description of both plants, camelina and canola, e.g. what is the difference between them, their Latin names, etc.
Why exactly these varieties (Canadian c., Australian c., AARI c., UAI c.) of camelina and canola were chosen for the study?
- The breeding method is described unsatisfactorily. I understand that plants grew under the open sky, not in a greenhouse. Was irrigation the only source of water for plants or were there any precipitation in this area? How was then the water stress applied? What does it mean “by skipping irrigation”? Was it abandoned completely or were the plants watered rarely? How often?
How was the foliar treatment performed? Was it in the kind of “shower”? Was not the foliar applied selenium solution washed from leaves by watering, either rain or irrigation? The sentence in l. 134-135 is not quite clear – does it mean that the foliar treatment was repeated every 25 days, starting from the moment when seedlings were 10 days old?
Were the same plants investigated in the second year of experiment? (Do camelina and canola grow for two years?) If not, how were the seeds prepared? Which seeds were chosen? Do Authors bred plants “foliar treated” from the seeds of plants foliar treated in the previous year and “seed Se priming” plants from the seeds of plants which were treated in the same way in the previous year?
There is also no information on the reagent used for preparation of selenium solution.
- Section 2.3.2.4 is unfinished
- Figures 1-5 are illegible. The letters are too small and there are too many bars in the graphs.
- Sections 3.2, 3.2.1, 3.2.2, 3.3 are the repetitions with the same pattern and the same sentences.
- Discussion should be rewritten. Part of it (l.307-319) should be rather moved to Introduction. In some sentences it is unclear whether the Authors describe their own results or those from references.
Reviewer 2 Report
In Ahmed et al.’s manuscript “Selenium alleviated the adverse effect of drought in camelina and canola through physio=biochemical alterations”, the authors found that selenium treatments affected yield components along with some physiological and biochemical traits. While detailed characterization of Se treatments for agriculture are worthwhile, I believe that better detailing of the experiments and analysis of the factors related to the changes in yield are needed before publication in Molecules is warranted.
In the Introduction, the authors should mention that Brassica species can accumulate Se and that this has been postulated to impact herbivory (Hanson et al 2004, New Phytologist, 162: 655). The potential role of differences in insect damage related to Se treatment should be addressed in the Results.
In the Methods, more details are needed on the experiment, including number or replicates, plant per replicate, plant spacing, timing of planting, and harvest, amounts of irrigation. Also the timing of measurements. In that regard, for the reproductive measurements, are the flowering times similar for the varieties or are measurements taken at different times?
In the Results, Table 1, it is not clear what is included as treatment, irrigation level as well as Se? This confusion is also propagated in the text.
In Section 3.1, the discussion of osmotic effects, canola tends to be larger in growth habit than camelina, which can explain the mentioned differences w/o being related to performance.
In Section 3.3, the large TSW of camelina, similar to larger than the canola, is surprising. In my experience camelina has been decidedly smaller seeded than canola, as has been described in the literature (e.g. Gugel and Falk 2006, Berti et al 2011, Enjalbert et al 2013).
From Figure 5, it is also not clear that there is an irrigation-level effect on yield in the control plants, which raises the question of the details of the treatment. I feel some of these results need to be presented differently to better illustrate the main results of the experiment.
In the Discussion, the narrow window between Se-deficiency and toxicity could be discussed in the context of these treatments.
In the Discussion, line 380, reference 75 is incorrect.